# Biomarkers in Ovarian Cancer: Towards Personalized Medicine

**DOI:** 10.3390/proteomes12010008

**Published:** 2024-03-18

**Authors:** Carlos López-Portugués, María Montes-Bayón, Paula Díez

**Affiliations:** 1Department of Physical and Analytical Chemistry, Faculty of Chemistry, University of Oviedo, 33006 Oviedo, Spain; uo269482@uniovi.es; 2Health Research Institute of the Principality of Asturias (ISPA), 33011 Oviedo, Spain

**Keywords:** ovarian cancer, diagnosis, prognosis, treatment response, biomarkers, cisplatin, targeted therapy, nanoparticles, proteomics

## Abstract

Ovarian cancer is one of the deadliest cancers in women. The lack of specific symptoms, especially at the initial stages of disease development, together with the malignancy heterogeneity, lower the life expectancy of patients. Aiming to improve survival rates, diagnostic and prognostic biomarkers are increasingly employed in clinics, providing gynecologists and oncologists with new tools to guide their treatment decisions. Despite the vast number of investigations, there is still an urgent need to discover more ovarian cancer subtype-specific markers which could further improve patient classification. To this end, high-throughput screening technologies, like mass spectrometry, are applied to deepen the tumoral cellular landscape and describe the malignant phenotypes. As for disease treatment, new targeted therapies, such as those based on PARP inhibitors, have shown great efficacy in destroying the tumoral cells. Likewise, drug-nanocarrier systems targeting the tumoral cells have exhibited promising results. In this narrative review, we summarize the latest achievements in the pursuit of biomarkers for ovarian cancer and recent anti-tumoral therapies.

## 1. Ovarian Cancer: A Rare but Fatal Malignancy

Ovarian cancer (OC) is the most fatal gynecological malignancy that affects the ovaries. However, the term OC also includes peritoneal and fallopian tube cancers since it may be unclear to define the tumoral origin. OC often shows chemoresistance, immunosuppression, and metastasis [1,2]. In worldwide terms, OC is becoming a burgeoning issue due to the increase in cases in the last few years [3], with almost 314,000 new patients in 2020 and a 32% increase in an eight-year time frame. In standardized terms, there is an incidence level of 6.6 cases per 100,000 people, with a mortality rate of 4.2 cases/100,000 people [4]. OC presents a ~45% five-year survival rate with a maximum incidence in the elderly female population (i.e., 70–85 years old) [5,6]. Factors such as lack of screening tools, late diagnosis caused by vague and nonspecific symptoms, and high rate of recurrence (70–80%) influence its poor prognosis [7,8]. Transvaginal ultrasonography and the detection of carbohydrate antigen 125 (CA-125) blood levels are the routinely used approaches in the clinical setting to diagnose OC [9,10]. However, these techniques lack enough specificity and often present artefacts that hamper the correct analysis of the results. This leads, in the end, to the abovementioned low patient survival rates [11].

### 1.1. Classification of Ovarian Carcinomas

The World Health Organisation classifies OC into two main groups: epithelial (95% of cases), and non-epithelial (including germ cell (<3%), sex-cord-stromal (<2%), and small cell (<1%) carcinomas) [7,12]. Within the epithelial category, there are five major histotypes: high-grade serous carcinoma (HGSC, ~70% of cases), clear cell carcinoma (~10%), endometrioid carcinoma (~10%), mucinous carcinoma (<5%), and low-grade serous carcinoma (LGSC, <5%) [13,14] (Figure 1A).

Also, OC can be subdivided into Type I and Type II based on the molecular and genetic features (Figure 1A and Table 1). While Type I often shows genomic alterations in *KRAS*, *BRAF*, *PTEN*, *PIK3CA*, *CTNNB1*, and *ARID1A* genes, they are genetically stable and the progression is slow. Type II is characterized by mutations in *TP53*, genetic instability, and rapid progression. Type I OC tumors can include LGSC, endometrioid, clear cell and mucinous carcinomas, whereas HGSC belongs to Type II tumors [15,16].

### 1.2. Strategies for the Treatment of Ovarian Cancer

Currently, diverse treatment options are available to treat ovarian tumorigenesis (Figure 1B):

Cytoreductive surgery. At the initial stages of tumor development, the most common procedure is the resection of the tumoral mass by laparotomy [31].Chemotherapy. For more advanced stages, the tumor resection procedure is often combined with chemotherapeutic approaches [31]. Among them, platinum-based treatments (cisplatin and carboplatin) in combination with paclitaxel are the first line of treatment for OC. This therapy has been applied during the last 20 years with no other treatment outperforming it [32]. Only bevacizumab, an anti-angiogenic drug, was introduced in 2011 to complement the platinum/paclitaxel combination [33,34]. Despite the high-rate effectiveness of this first-line treatment, any therapeutic improvements are still welcome to improve drug outcomes, being drug transport efficiencies the most important limiting factors in existing treatments. In this regard, hyperthermic intraperitoneal chemotherapy (HIPEC) allows the single administration of high doses of the cytostatic while also exploiting the effect of hyperthermia (41–43 °C for 30–120 min), improving the drug cytotoxicity. The clinical trial OVHIPEC-1, performed in The Netherlands and Belgium, showed higher disease-free survival and overall survival rates in the patients undergoing surgery resection and HIPEC vs. those who underwent surgery alone [35,36,37]. Likewise, the use of drug nanocarriers appears as a promising alternative to ensure the successful delivery of drug-based treatments [38,39,40]. This option will be further discussed in Section 4.Immunotherapy. New immune-based therapies are also under investigation for OC. Thus, the inhibition of specific proteins (like PD-1) by drugs (e.g., nivolumab) that results in the promotion of anti-tumor immunity is showing promising results in the field, with ~15% of OC cases positively responding to the treatment [41]. Other therapies using immune modulators or immune checkpoint inhibitors are also applied to patients.Targeted therapies. In 2014, poly(ADP-ribose) polymerase (PARP) inhibitors were approved as maintenance therapy for patients with recurrent disease after platinum treatments. PARP inhibition leads to the accumulation of double-strand breaks that cannot be repaired in cells that are homologous recombination repair deficient (HRD), finally leading to cell death. Considering around 50% of HGSC tumors are HRD, this therapy has been reported as an important alternative [42]. Three clinical trials in phase III showed promising results leading to the approval of niraparib [43], olaparib [44], and rucaparib [45] drugs. Recently, PARP inhibitors are also under evaluation in the front-line setting (rather than maintenance therapy) via four phase-III clinical trials [46,47,48,49].Other targeted therapies include the inhibition of proteins of the tropomyosin receptor kinase (TRK) family. The binding of neurotrophins to TRK receptors activates Ras, PI3K and phospholipase C-γ1 signaling cascades in a normal state. However, any rearrangement of these receptors may lead to cell malfunctioning and tumorigenesis due to overactivation of signal transduction [50].Hormonal therapies. Since OC progression depends on hormones released from the hypothalamic-pituitary-ovarian axis and considering the demonstrated efficacy of hormone therapies in breast and endometrial cancers, these therapeutic strategies have been stated for the treatment of patients showing platinum resistance and tumor recurrences. While gonadotropins, estrogens, and androgens promote OC advancement, gonadotropin-releasing hormone (GnRH) and progesterone might have a protective role [51]. Thus, analogues of GnRH (e.g., triptorelin), or inhibitors of estrogen (e.g., tamoxifen) and androgen (e.g., flutamide), are used in the clinics [52].

Of note, radiotherapy (ionizing radiation) is rarely used for the treatment of OC due to its high toxicity rates and lack of specificity. However, it might be applied to help symptoms (for pain control) of advanced stages [53].

Altogether, more research about OC, its diagnosis, prognosis, and treatment options are still required to improve the life expectancies of women suffering from this fatal disease. All these aspects will be covered in the following sections.

## 2. Biomarkers in Ovarian Cancer: From Diagnosis to Prognosis

Since OC lacks remarkable symptoms at the initial stages of the disease, investigations have been oriented towards the identification of effective diagnostic biomarkers. The primary objective of these studies is to discern distinctive proteins and molecules intricately linked to the initiation of OC [54]. These are present in blood serum and plasma, which are low invasive and easily accessible samples. Hitherto, more than fifteen markers have been recognized. Within the serum markers, CA-125, HE4, kallikreins, prostasin (PSN), transthyretin (TTR), transferrin, and vascular endothelial growth factor (VEGF) are worth mentioning. As for the plasma markers, apolipoprotein A-I (apoA-I) and osteopontin (OPN) are commonly screened in the clinics. A summary of these diagnostic biomarkers is shown in Table 2.

Moreover, the genetic landscape of OC is explored through the examination of mutations in genes such as *BRCA1* and *BRCA2* (Breast Cancer genes 1 and 2). These tumor suppressor genes are involved in DNA repair, cell cycle regulation, and genomic stability, preventing the formation of abnormal cells and therefore, the development of certain cancers, particularly breast and OC [55]. Germline mutations in these genes are related to a higher risk of developing OC (39–44% cumulative life risk by age 70 when *BRCA1* is mutated, and 11–17% when *BRCA2* is mutated, vs. 1–2% risk in the general population) [56]. Also, it has been observed that those patients presenting germline *BRCA1* and *BRCA2* pathogenic variants show a better response to platinum-based treatments [57]. As for the somatic mutations, they are detected in 5–7% of OC cases [58]. Of note, better short-term survival rates have been observed in those patients carrying *BRCA* mutations; however, there are some contradictory observations at longer times, with studies reporting conflicting mortality rates for BRCA-mutated carriers vs. noncarriers [59,60,61].

Prognostic factors are also key players in OC due to the abovementioned late diagnosis of the disease. These markers are found in serum and plasma, and some of them are also used in diagnosis (e.g., CA-125, OPN and VEGF). Others like bikunin and creatine kinase B (CKB) only predict cancer development (Table 2).

**Table 2 proteomes-12-00008-t002:** Selection of diagnostic and prognostic markers usually screened in the clinics.

	Biomarker	Full Name	Features	Specificity/ Sensitivity	Diagnostic/ Prognostic Marker?	References
Serum markers	CA-125	Carbohydrate antigen 125	- Highly present in 80% of late-stage epithelial OC - Present in other non-tumoral conditions (e.g., endometriosis, normal menstruation, pregnancy) → no longer recommended for screening and diagnosis	90%/60%	yes/yes	[7,62,63,64,65,66]
	HE4	Human epididymis protein	- Expressed in endometrioid and serous OC - Present in some postmenopausal conditions	95%/73%	yes/no	[62,63,67,68,69]
	KLK	Kallikrein	Upregulated in OC (serum and ascites) with poor prognosis and chemoresistance to paclitaxel	75%/77%	yes/no	[62,70]
	PSN	Prostasin	Expression levels > 100x in epithelial and stromal OC vs. normal condition	94%/51%	yes/no	[62,71]
	TTR	Transthyretin	Low levels in OC	69%/79%	yes/no	[62,72]
	Transferrin	Transferrin	Low levels in OC	74%/73%	yes/no	[62]
	VEGF	Vascular endothelial growth factor	Direct correlation with OC	74%/79%	yes/yes	[7,62]
	Bikunin	-	High levels related to favorable prognosis	70%/75%	no/yes	[62,73]
	CKB	Creatine kinase B	Highly expressed in early tumoral phases	94%/92%	es/yes	[62,74]
Plasma markers	apoA-I	Apolipoprotein A-I	Low levels in OC	98%/94%	yes/no	[62,75]
	OPN	Osteopontin	Highly expressed in OC	34%/81%	yes/yes	[62,76]

*OC*, ovarian cancer.

Despite the number of markers available for OC diagnosis and prognosis, most of them do not show sufficiently high sensitivity or specificity on their own (Table 2). However, their combined analysis has reported a stronger discrimination capability (e.g., TTR + apoA-I + transferrin + CA-125; CA-125 + apoA-I + β2-microglobulin; PSN + CA-125; IGFBP2 + LCAT + CA-125 [77]; CA-125 + CA 19–9 + EGFR + G-CSF + eotaxin + IL-2R + cVCAM + MIF [78]; CA-125 + vitamin K-dependent protein Z + phosphatidylcholine sterol acyltransferase + C-reactive protein [79]), as various algorithms have shown. In this regard, OVA1 is a multivariate index assay that combines the levels of five biomarkers (CA-125, TTR, apoA-I, β2-microglobulin, and transferrin) to assess the likelihood of malignancy in pelvic masses. It provides a risk stratification algorithm that categorizes patients into low-, moderate-, or high-risk groups based on the biomarker panel results, improving the sensitivity of OC detection compared to clinical and radiological assessments alone [80]. Another example is ROMA (Risk of Ovarian Malignancy Algorithm). ROMA uses a mathematical algorithm that combines CA-125 and HE4 levels along with menopausal status to improve the accuracy of OC risk assessment. As in the OVA1, this combination of factors helps categorize patients into low-risk or high-risk groups for ovarian malignancy, consolidating a supplementary test to assist clinicians in determining the likelihood of cancer before surgery [81,82]. Lastly, it is also remarkable the existence of two more algorithms based on the CA-125 biomarker alone. The first one is the CA-125 Risk of Ovarian Malignancy Algorithm (ROCA), which utilizes serial CA-125 measurements over time to identify rising trends that may indicate the presence of OC. The second one is the Risk of Malignancy Index (RMI), which, in a similar way as ROCA, is based on CA-125 levels but also considers patient-reported symptoms, aiming to enhance the accuracy of OC detection, especially in postmenopausal women [83].

On top of these serum and plasma markers, new investigations are oriented towards other molecules (e.g., mRNA, lncRNA, DNA) present in, e.g., blood and cells [84]. Moreover, other proteins are emerging as promising candidates for predicting the diagnosis and prognosis of OC. For instance, aryl-hydrocarbon-receptor-nuclear-translocator-like (ARNTL), runt-related transcription factor 3 (RUNX3), calstabin-1/paired box protein 9 (Fkbp1/Pax9), collagen alpha-1(XI) chain (COL11A1), forkhead box protein M1 (FOXM1), retinoblastoma-binding protein (RBP4), proliferation marker protein (Ki-67), aldehyde dehydrogenase 1 (ALDH1), folate receptor alpha (FOLR1), and glutathione S-transferase polymorphisms (GSTP), among others [62,85,86].

Nevertheless, it is also important to note that the clinical utility of all these biomarkers and algorithms varies, and their use may be influenced by factors such as cancer stage, histological subtype, and individual patient characteristics. Also, as the field of OC biomarker research continues to evolve, the integration of multiple markers and advanced analytical approaches holds promise for more precise diagnosis, prognosis, and the development of targeted therapeutic strategies. This ongoing exploration of biomarkers aims to contribute significantly to the early detection of the disease and the development of patient-personalized treatments.

## 3. Mass Spectrometry-Based Proteomics Studies in Ovarian Cancer

The proteomics field has an important role in the scientific research of OC, facilitating high-throughput investigations via sophisticated analytical methodologies such as mass spectrometry (MS). These approaches afford a comprehensive scrutiny of the proteomic complexity intrinsic to OC cells, allowing for the discovery of novel biomarkers and the comprehensive characterization of this malignancy. The knowledge of what proteins, signaling pathways, and even metabolites are altered in OC and its different subtypes could undoubtedly help to assist with diagnosis, guide treatment choices, and improve quality of life and survival expectancies for women suffering from this pathology.

To understand to what extent the OC is investigated using high-throughput proteomics strategies, a thorough in silico research in the PRIDE repository (https://www.ebi.ac.uk/pride/; accessed on 1 November 2023) using the term “ovarian cancer” was done reporting 170 studies (December 2023) related to this topic. A summary of the most relevant investigations is mentioned below and in Table 3.

Ahn et al. [87] performed a metabolomic and proteomic investigation on peripheral blood from HGSC patients, identifying more than 1200 proteins and almost 400 metabolites. The latter was distributed into polar lipids (51% of total; including 63 phosphatidylcholines, 9 lysophosphatidylcholines, and 4 ceramides), small molecules (14%; 20 amino acids, 7 biogenic amines, 1 monosaccharide) and neutral lipids (35%; 16 acylcarnitines, 13 diglycerides, 30 triglycerides, 25 sphingomyelins, and 10 cholesteryl esters). And 34 of them showed upregulated levels in the healthy control samples. As for the proteome, 197 differentially expressed proteins were detected in the OC group (108 upregulated, 89 downregulated; vs. the healthy group). The OC-upregulated plasma proteins were involved in platelet, immune system, gluconeogenesis, homeostasis, extracellular matrix, response to stimuli, and signaling functions. All these reflect the active energy metabolism, cancer growth, and role of the cancer environment in OC development.

Aiming at discovering drivers of long-term survival, Formalin-Fixed Paraffin-Embedded (FFPE) resistant and sensitive HGSC samples were profiled, and CT45 protein was pinpointed as an independent prognostic factor for OC, associated with a two-times increased cancer-free time [88]. Further research reported a cytotoxic-T-cell-derived mechanism in which the CT45-derived HLA-I peptides are recognized by the T-cell receptor activating tumor cell death.

To further study the heterogeneity of HGSC, a phosphoproteomics analysis was performed on 30 patient-derived tumor samples undergoing complete tumor resection or neoadjuvant chemotherapy [89]. The investigation resulted in 101 differentially expressed proteins related to endocytosis, engulfment and cell spreading, among others, and 71 significantly altered proteoforms between the two groups, showing different phosphorylation patterns. Of note, neurofibromin 1 (NF1) was remarkably less expressed in the group undergoing resection (also verified by transcriptomics and immunohistochemistry techniques). NF1 has been described to have a role in the RAS/MAPK signaling pathway, tumorigenesis, and chemotherapy resistance in HGSC.

Also focusing on protein modifications, McGee and collaborators [90] introduced a new method (AutoPiMS) for the identification of proteoforms in HGSC. This platform is based on a semi-automated top-down proteomics approach performed directly from tissues. It offers information about the molecular mass of the proteoform, its spatial location, and quantitative value. They applied it for the analysis of tumor and stromal ovarian cancer biopsies, identifying ~1000 proteoforms, 303 of which were differentially expressed between the two sample types. Similarly, a MALDI-MS imaging procedure combined with top-down proteomics allowed for the identification of 15 novel truncated proteoforms (alternative proteins) distinguishing between tumor and benign cells of serous OC [91].

Another proteomics study focused on the comparison between OC and normal ovarian tissue (using 11 paired biopsies) revealing more than 2000 altered proteins, with a significant role in mitochondrial proteostasis and protein translation [92]. Specifically, HSP60 was found to be highly upregulated in OC. Similarly, research on OC plasma revealed that SPARC and THBS1 proteins are present in elevated concentrations (compared to healthy donors) [93]. As for the epithelial OC, a study from 2019 using the SKOV3WT cell line reported Septin-2 as a potential target for this OC subtype, since knocking down this protein resulted in a tumor proliferation decrease [94].

**Table 3 proteomes-12-00008-t003:** Summary of mass spectrometry-based studies on ovarian carcinoma discussed in this review.

Sample Type	Sample Origin	OC Subtype	Studied Analytes	MS Technology	Outcome Summary	Reference
Tumor tissue	Patients (25 cases) and cell lines	HGSC	Proteins and phosphoproteins	LC-MS/MS	8190 quantified proteins	[88]
	Patients (103 cases)	Mesenchymal HGSC	Proteins	SWATH/DIA-MS and iTRAQ-DDA	4363 by iTRAQ-DDA and 1659 by SWATH/DIA-MS (1599 in common)	[95]
	Patients (30 cases)	HGSC	Proteins and phosphoproteins	TMT-based LC-MS/MS	7290 proteins and 12,914 phosphosites	[88,89]
	Patients (11-paired normal and tumoral cases)	Serous, clear cell, endometrioid carcinomas	Proteins	TMT-based LC-MS/MS	7719 proteins	[92]
	Patients (20 cases)	HGSC and endometrioid carcinoma	Proteins	LC-MS/MS	8-marker panel for discrimination between HGSOC and endometrioid carcinoma	[96]
	Patients (31 cases)	Serous OC	Proteins	MALDI imaging MS	3844 proteins	[97]
Blood	Patients (20 cases)	HGSC	Plasma metabolites and proteins	Nano-LC-ESI–MS/MS and MRM-MS	34 metabolites (L-carnitine and PC-O) and 197 proteins (PPCS, PMP2, and TUBB)	[87]
Ascites	Patients (70 cases)	HGSC	Macrophage secretome	LC-MS/MS	Focus on TGFB1, TNC and FN1 (low levels relate to better survival rates)	[98]
Cell culture	Patient (2 patient-derived primary cell lines)	HGSC	Proteins and phosphoproteins	LC-MS/MS	4151 quantified proteins, and 2905 phosphorylation sites	[99]
	Patients (8 cases) and cell lines (30)	HGSC	Proteins	LC-MS/MS	>10,000 proteins (67-protein signature)	[100]
	Cell line (SKOV3WT)	Serous and clear-cell OC	Proteins	LC-MS/MS	Septin-2 as protein target to reduce tumorigenesis	[94]
	Cell line (OVCAR-3, SKOV-3)	HGSC and non-serous OC	Proteins and phosphoproteins	LC-MS/MS	3324 proteins, 2978 phosphopeptides	[101]
	Cell lines (8)	Epithelial OC	ECM1-interacting proteins	LC-MS/MS	ECM1a, integrin aXb2, hnRNPLL, and ABCG1 as potential targets	[102]

*DDA*, data-dependent acquisition; *DIA*, data-independent acquisition; *ECM1*, extracellular matrix protein 1; *ESI*, electrospray ionization; *HGSC*, high-grade serous ovarian carcinoma; *iTRAQ*, isobaric tag for relative and absolute quantitation; *LC*, liquid chromatography; *MALDI*, matrix-assisted laser desorption/ionization; *MRM*, multiple reaction monitoring; MS/*MS*, tandem mass spectrometry; *OC*, ovarian cancer; *SWATH*, sequential window acquisition of all theoretical mass spectra; *TMT*, tandem mass tag.

Interestingly, Dieters-Castator et al. [96] defined an 8-marker panel (KIAA1324, PAM, PGR, WT1, SCGB2A1, PIGR, CTNNB1, TP53) for discriminating endometrioid carcinoma from HGSC using freshly frozen tumor samples. The selection was made from >500 proteins differentially expressed between the two conditions, of which 106 were sufficient to rightly distinguish 90% of samples. Supplementary validation identified KIAA1324 as a highly discerning biomarker for endometrioid carcinoma.

Coscia et al. [100] performed a large study on different OC cell lines and HGSC tumor samples reporting > 10,000 proteins (8397 (77%) in common between all samples) which allowed for the definition of a 67-protein signature to classify OC samples originated either by the ovarian surface epithelium or fallopian tube epithelial cells.

Finally, Nguyen and colleagues [99] studied the effect of cisplatin treatment, and the frequently derived drug resistance, in the proteomes and phosphoproteomes of two patient-derived cell lines (cisPt sensitive and resistant cell lines). Their screening reported significant differences, showing elevated levels of phosphorylated sequestosome-1 proteoform in the resistant condition. The authors suggested this protein, linked to the downregulation of apoptosis, as a potential marker for drug-resistance change in HGSC cells.

Overall, more than 11,000 distinct proteins were identified across all studies included in this review. The investigation of Coscia et al. [100] on 30 OC cell lines reported a great coverage of the tumoral ovarian proteome, with 8682 proteins in common with the patient-derived data (Figure 2A). Remarkably, 1525 proteins were only detected in the study of Coscia et al. (Figure 2B), demonstrating the strength of including a large variety of cell models to profile a tumoral condition. Nevertheless, different protein profiles were described per individual cell lines, highlighting the limitations of using cell lines to thoroughly characterize the OC. Thus, studies employing only one or two cell line models might have poor resolution and caution must be taken when interpreting that data. This has been evidenced when comparing the study from Nguyen et al. [99], performed in 2 cell lines, vs. the analysis of Coscia et al. [100] in 30 cell lines, with >5000 proteins uniquely identified in the latter manuscript. Likewise, investigations on patient samples must be carefully designed to avoid biased conclusions due to the wrong selection of cases (e.g., highly heterogeneous patient groups, or a small number of cases per group of study). For instance, Guo et al. [92] employed 11-paired normal and tumoral samples to specifically study the HSP60-regulated mitochondrial proteostasis. Despite the relevance of their findings, the selected OC samples included two endometrioid, four clear-cell, and five serous carcinomas, being a reduced and heterogenous group of samples that might incompletely represent the tumor phenotype. Nevertheless, patient-derived samples are preferred over cell line-based studies, since the latter might poorly resemble the OC phenotype, as previously mentioned. Moreover, despite the relevance of high-throughput proteomics studies, biological interpretation of the data should always be discussed to avoid just reporting large lists of proteins without any established causative pathway.

Finally, the evaluation of the quantitative expression of two proteins (Septin-2 and HSP60) across different studies in cell lines and patients was assessed (Figure 2C). Overall, cell line-derived data obtained within the same study [100] reported comparable results for both proteins across all 30 OC cell lines analyzed. However, another study performed on the SKOV3WT cell line [94] showed great differences, depicting higher levels of HPS60 and a lower expression degree of Septin-2 compared to the other studies. This is particularly noteworthy considering this last study was focused on determining the role and expression of Septin-2 in OC vs. benign conditions. Finally, patient-derived data depicted mixed results, proving again the importance of sample cohort selection.

## 4. New Therapeutic Strategies in OC Based on Drug-Nanodelivery Systems

As mentioned earlier in this review, the current first-line treatment option for OC is based on the combination of surgical resection and chemotherapy, with the latter mainly including platinum-based antitumoral drugs. These treatments are often related to the development of drug resistance, hampering the elimination of the cancer cells, and diminishing the survival chances of treated patients. Thus, new therapeutic strategies in OC, integrating the usage of biomarkers and nanodelivery systems, represent a groundbreaking frontier in personalized and targeted medicine, enhancing the precision, efficacy, and safety of OC treatments [103]. On the one hand, as previously indicated, biomarkers play a pivotal role in tailoring therapeutic strategies by providing valuable insights into the disease’s characteristics, thus guiding treatment decisions [104]. On the other hand, nanocarriers offer mechanisms for targeted drug delivery via surface modification with ligands or molecules (e.g., antibodies) to specifically bind to receptors present in OC cells [105]. Thus, drug delivery precision is enhanced, maximizing its therapeutic impact on malignant cells while minimizing its exposure to healthy tissues.

Among the wide range of available nanocarriers, liposomes, nanoparticles, micelles, and dendrimers are worthy to be mentioned (Table 4). All these nanodelivery systems offer a sophisticated platform for targeted drug delivery, enabling the encapsulation of therapeutic agents, and thus addressing challenges such as drug solubility, stability, and side effects [106,107].

Liposomes. These vesicles composed of lipid bilayers present the ability to encapsulate both hydrophobic and hydrophilic substances, making them particularly adaptable for the targeted delivery of therapeutic agents. Liposomes provide a protective environment, shielding the drugs from degradation and ensuring their stability [108]. This property is especially advantageous in OC treatment, where the delivery of chemotherapeutic drugs is critical for effective tumor regress [109]. Likewise, liposomal formulations of drugs such as doxorubicin and paclitaxel have been developed to address challenges related to drug solubility, bioavailability, and toxicity [110,111]. Some examples have gained clinical approval for OC, including Doxil and Lipo-PTX, liposomal formulations of doxorubicin and paclitaxel, respectively [112,113,114]. These formulations have shown promising outcomes in clinical settings, offering prolonged circulation times, reduced toxicities, and improved therapeutic indices compared to their conventional counterparts. Another clinical study evaluated the combined use of paclitaxel liposomes and carboplatin with the administration of the free drugs, reporting a significantly enhanced response in the encapsulated condition with reduced side effects [115]. Furthermore, one notable benefit of using liposomes is the reduction of systemic side effects associated with chemotherapy. By facilitating targeted drug delivery, liposomes minimize the exposure of healthy tissues to potent chemotherapeutic agents, leading to a more favorable safety profile. Likewise, by targeting overexpressed receptors (e.g., luteinizing hormone-releasing hormone receptor, LHRHR), liposomes increase their uptake rate, enhancing cell apoptosis, as shown in in vitro studies in A2780 cells [116]. However, the use of liposomes in OC therapy presents some stability issues and the transition from laboratory-scale to large-scale production presents obstacles that require ongoing research and technological advancements [117].

**Table 4 proteomes-12-00008-t004:** Summary of nanocarrier platforms, their benefits, and drawbacks.

Nanocarrier	Features	Advantages	Disadvantages	Examples	References
Liposome	Encapsulation of hydrophobic and hydrophilic substances	- biocompatibility - surface modification - reduction of side effects	- stability issues - hard transition to large-scale production	Doxil, Lipo-PTX	[112,113,114]
Nanoparticle	Delivery of drugs attached to its surface or by encapsulation	- easy synthesis - size control - surface modification	- heterogeneous synthesis processes - concentration-dependent toxicity for patients	Cis-platin coated iron nanoparticles, gold nanoparticles, albumin-based nanoparticles	[38,118,119,120,121,122]
Micelle	Encapsulation of hydrophobic drugs	- biocompatibility - drug stabilization - surface modification	- difficult delivery of hydrophilic substances	Genexol-PM, PEG-based micelles, poly(propylene oxide) (PPO)-based micelles	[123,124]
Dendrimer	Functionalization of its dendritic architecture with ligands for targeted drug delivery	- co-delivery of substances - surface modification	- heterogeneous - difficult synthesis processes	Phosphorus (P-dendrimers), polyamidoamines (PAMAM), polypeptides, polyesters	[125,126,127,128]

Dendrimers. These nanocarriers are highly branched macromolecules with well-defined structures that offer the potential for targeted drug delivery, and that can be functionalized with ligands for specific interactions with cancer cells [126,127]. Their dendritic architecture allows for precise control over drug loading, enabling the encapsulation of therapeutic agents within their well-defined branches. Like liposomes, dendrimers can also be modified by attaching ligands or antibodies to their surface, improving their specific targeting [129]. This dendritic structure also allows the co-delivery of multiple drugs with different functionalities, enabling synergistic effects and overcoming drug resistance mechanisms often encountered in conventional OC treatment. This aspect makes dendrimers valuable tools for designing personalized therapeutic approaches tailored to the specific characteristics of each cancer case [125]. Several in vitro studies have reported the benefits of using drug-coupled dendrimers. For instance, its combination with cisplatin to treat OVCAR3, SKOV, A2780, and CP70 cells reported a 7-x increase in the expression of apoptotic genes and a 2-x increase in the activity of caspases, ultimately leading to tumoral death [130].Nanoparticles. Most nanoparticles used in OC belong to either polymeric, metallic, or albumin-based nanoparticles. Polymeric nanoparticles are constructed from biodegradable polymers like poly(lactic-co-glycolic acid) (PLGA) or polyethylene glycol (PEG) [118,119], while metallic nanoparticles are made of metallic elements such as gold, silver, or iron [120,121,131], and albumin-based nanoparticles, like Abraxane, use albumin aggregates as a carrier. Their versatility allows the controlled release of drugs like paclitaxel, olaparib, or cisplatin, and they can also get their surface modified to allocate specific targeting molecules [38,46,107,132,133]. Moreover, nanoparticles can encapsulate therapeutic agents, preventing premature drug degradation and ensuring their sustained release [134,135]. Also, the small size of nanoparticles contributes to their ability to passively target tumors through the Enhanced Permeability and Retention (EPR) effect. This phenomenon leverages the leaky vasculature surrounding tumors, allowing nanoparticles to accumulate selectively in cancerous tissues. The passive targeting mechanism enhances drug delivery efficiency and ensures a higher concentration of therapeutic agents at the tumor site [136]. As for albumin-based nanoparticles, they benefit from the natural affinity for the albumin receptor on cancer cells, facilitating targeted drug delivery. This approach improves drug solubility, reduces the need for toxic solvents, and enhances the therapeutic effects of drugs like paclitaxel in OC. Additionally, their biocompatibility is a critical factor in minimizing systemic toxicities associated with chemotherapy, as it presents a benefit from the natural origin of this protein, which normally is well-tolerated by the body, reducing the risk of adverse reactions [122,137]. Nanoparticles have been evaluated in multiple in vitro studies showing promising results. For instance, PLGA-based nanoparticles carrying molecules to specifically bind the LHRH receptor (i.e., LHRH-a) and delivering CPT-11, an inhibitor of DNA topoisomerase I, significantly inhibited the cellular proliferation of A2780 cisplatin-resistant cells [138]. Also, degradable mesoporous silica nanoparticles encapsulating paclitaxel showed enhanced toxicity in OVACAR-3 and PA-1 cells [139]. Another investigation in vitro studied the role of the nanoparticle surface charge, reporting that nonionic polymeric nanostructures reduce cancer cell viability at greater levels compared to positively charged formulations [140]. Despite the promising results of these nanostructures, they are not routinely applied for the treatment of OC patients.Micelles. These nanostructures are formed by the self-assembly of amphiphilic molecules in aqueous solutions and offer a multifaceted approach to addressing key challenges associated with traditional drug delivery [124]. One of their distinctive features is their biocompatibility which contributes to their potential for minimizing systemic toxicities associated with chemotherapies [123]. They also present the ability to encapsulate hydrophobic drugs within their core [141] and to include modifications in their surface [106,142]. Micelles, together with liposomes, are the only nanostructures approved by a national drug administration to be used in patients. Specifically, in 2007, a paclitaxel-carrying PEG-PLA polymeric micelle (Genexol-PM) was approved in South Korea for breast, lung, and OC treatment [143,144]. Other investigations have also shown promising results in vivo and in vitro, such as the encapsulation of paclitaxel in micelles with epidermal growth factor (EGF) as targeting molecule that showed an improved uptake by SKOV3 cells subsequently inhibiting their proliferation [145].

### Surface Modification of NP to Promote Targeted Active Drug Uptake

The active targeting of nanocarriers is performed via the surface inclusion of tumoral-specific ligands that specifically identify and target cancer cells while minimizing damage to normal counterparts. Among the OC molecular targets, it can be mentioned the Transferrin Receptor 2 (TFR2), the AXL receptor, the VEGF receptor (VEGFR), and the folate receptor.

TFR2. It is a transmembrane glycoprotein that plays a pivotal role in the regulation of iron homeostasis within the human body thanks to its interaction with transferrin, a protein responsible for transporting iron in the bloodstream, which facilitates the sensing of iron levels. In this sense, OC cells often exhibit alterations in iron homeostasis to support their rapid proliferation and growth [146,147,148]. Possibly due to this effect, this receptor is overexpressed in some OC cell lines, making it a suitable molecule for targeted therapies [149].AXL receptor. It is a member of the family of receptor tyrosine kinases alongside Tyro3 and Mer [150]. AXL is frequently overexpressed in OC cells, and its upregulation has been associated with aggressive tumor behavior, metastasis, and resistance to conventional therapies. The activation of AXL signaling pathways contributes to processes such as epithelial-to-mesenchymal transition (EMT), which enhances the invasive potential of cancer cells. Moreover, AXL has been involved in immune evasion, dampening the antitumor immune response. This receptor’s role in promoting cell survival and inhibiting apoptosis further underscores its significance in OC progression. Thus, the AXL receptor can be employed dually: (i) exploring AXL inhibitors as potential therapeutic agents to counteract the aggressive features of OC [151,152], and (ii) targeting AXL receptor within nanodelivery systems to improve drug incorporation and release efficacies.VEGFR. Related to the metastasis field, the receptor for the angiogenic VEGF factor, a biomarker described in Section 2, has become a key strategy in the development of targeted therapies for OC [153,154]. Anti-angiogenic drugs, such as bevacizumab, an anti-VEGF monoclonal antibody, specifically target VEGF or its receptor to block the formation of new blood vessels, thus restricting the access of oxygen and nutrients to the tumor, thus making bevacizumab an excellent candidate for developing new targeted therapies [47,155,156,157,158]. Moreover, the receptor could be used as a potential target for drug delivery.Folate receptor. The overexpression of this receptor on OC is often associated with increased tumor aggressiveness, and poor prognosis and can be leveraged by the specific binding affinity to some drugs like mirvetuximab soravtansine-gynx, an antibody-drug conjugate designed to selectively deliver a chemotherapy agent to cancer cells that overexpress this receptor. This targeted approach aims to enhance the efficacy of chemotherapy while minimizing damage to healthy cells [159,160].Others. Multiple other receptors can be targeted to enhance treatments. For instance, the follicle-stimulating hormone receptor (FSHR) that is overexpressed in OC cells can be targeted by including the binding peptide domain of FSH (FSH33) onto the nanostructure surface, like dendrimers, exhibiting an increased tumoral selectivity [161]. Likewise, biotin functionalization of the surface might enhance the biotin receptor-mediated endocytosis uptake of nanosystems, as demonstrated in an in vitro study with OVCAR3 cells by Yellepeddi et al. [162].

Finally, nanodelivery systems can be engineered with imaging agents, allowing real-time visualization of drug distribution and tumor response with the help of artificial intelligence tools. This integration contributes to the development of theragnostic approaches, combining diagnostics and therapeutics in a single system [163].

All things considered, the customized combination of specific tumor-targeting biomarkers and nanodelivery systems facilitates tailored treatment strategies for OC. By previously characterizing the biomarker profile of each OC subtype or even each OC patient, such drug-target molecule-nanocarrier customization might ensure the selection of the optimal treatment in each case (personalized medicine). Advanced techniques available in research and clinical settings, like spectral flow cytometry and mass cytometry, would help in the definition of such biomarker profiles since they allow for the simultaneous characterization of >40 markers in a fast and reliable manner [164,165]. Therefore, it seems clear that the future of OC therapy lies in the continued exploration and refinement of integrated strategies. As research delves deeper into the molecular intricacies of OC subtypes and develops novel nanodelivery systems, the synergy between biomarkers and nanotechnology promises to redefine treatment paradigms. Personalized, targeted, and combination therapies guided by biomarkers and delivered by advanced nanocarrier systems are at the forefront of revolutionizing OC care.

## 5. Conclusions and Future Perspectives

As stated throughout this review, the research field of OC is yet to be completed. Remarkable efforts have been performed to fully understand the pathobiology of this malignancy and to discover molecules with a diagnostic and predictive role. The rise of high-throughput strategies based on genomics, transcriptomics, and proteomics has accelerated the characterization of the OC, providing researchers and clinicians with new knowledge for the ultimate development and selection of therapies. In this regard, proteomics-based approaches are key players since they offer information about the final effector molecules in the cells, the proteins. Thus, techniques such as MS have allowed the identification of biomarkers for diagnosis and prognosis as well as new potential tumoral targets. Nevertheless, these investigations often lack a deep analysis of the obtained results, and only report long lists of proteins without referring to any causative pathway or cellular/molecular mechanism. Occasionally, these studies report contradictory data about specific markers, highlighting the need of meta-analyses to compare investigations. Furthermore, despite the advantages of using a cell line model, outcomes cannot be directly translated into the clinics and caution might be taken to avoid over-interpretation of the existing literature. More efforts should be oriented to either analyze larger series of cell lines to better represent the malignancy or include patient samples. As for this, the experiment design is also key, to make sure that the size and heterogeneity of the sample cohort are appropriate.

As for the treatment options for OC, it has been shown that several alternatives are available nowadays, with some of them displaying promising results. In this regard, it is remarkable the improvement of traditional therapies, like the HIPEC technology for administration of chemotherapeutic drugs, or the development of new approaches, like the use of specific inhibitors (e.g., PARP inhibitors). However, more investigations are still needed, which requires the collaborative effort of multidisciplinary groups including experts in bioanalytical chemistry, biotechnology, oncology, and medicine. Moreover, state-of-the-art technologies like mass cytometry or spectral flow cytometry might also be highly valuable to assist in the understanding of OC. Therefore, although there are still knowledge gaps to be filled, there is a promising perspective in the field towards a personalized view of the disease which will allow us to better diagnose and treat women suffering from this fatal malignancy.

## Figures and Tables

**Figure 1 proteomes-12-00008-f001:**
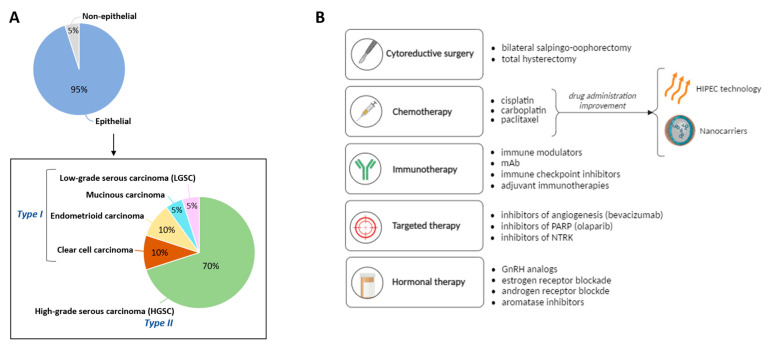
Classification of ovarian carcinomas (**A**) and therapeutic approaches commonly used in the clinics (**B**). *PARP*, poly(ADP-ribose) polymerase; *NTRK*, neurotrophic receptor tyrosine kinase; *GnRH*, gonadotropin-releasing hormone.

**Figure 2 proteomes-12-00008-f002:**
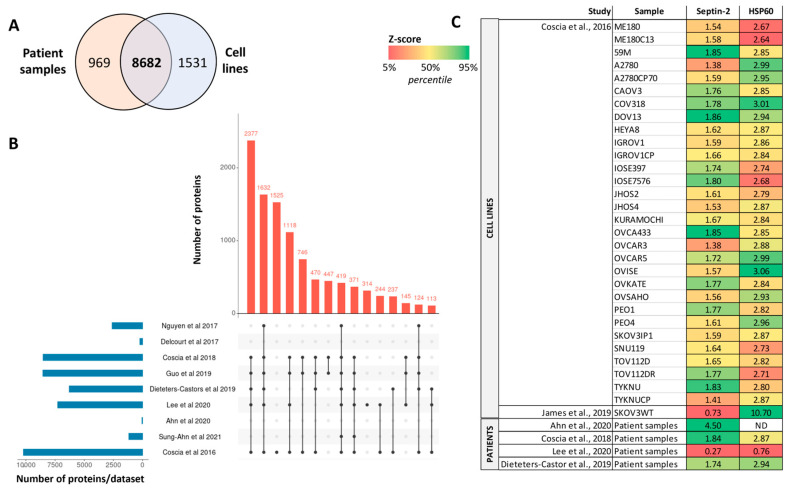
Meta-analysis of proteomics studies on ovarian cancer. (**A**) Intersection of identified proteins across studies using cell lines [94,100] and patient samples [87,88,89,91,92,93,96,99]. (**B**) UpSet plot depicting the common proteins identified between the described proteomics subsets (only intersection sets with >100 proteins are shown). (**C**) Quantification of Septin-2 and HSP60 proteins across studies. Z-scores were calculated for each protein to correct for different quantification scales. Color codes correspond to proteins in the 5% percentile (red) as top low values and 95% percentile (green) as top high values.

**Table 1 proteomes-12-00008-t001:** Features of ovarian carcinomas depending on their molecular, genetic and histological classification.

Molecular and Genetic Classification	Type I *Mutations in KRAS, BRAF, PTEN, PIK3CA, CTNNB1, ARID1A Genes,* *Genetically Stable and Slow Progression*				Type II *Mutations in TP53,* *Genetically Unstable and Rapid Progression*
Histotype	Clear Cell [17,18,19]	Endometrioid [17,20,21,22]	Mucinous [17,23,24,25,26]	LGSC [27]	HGSC [28,29,30]
Tumoral/cellular structure	Clear or hobnail-shaped cells with abundant cytoplasm, often containing glycogen and lipid droplets.	Glandular structures resembling those of the endometrium.	Glandular structures filled with mucin-producing cells.	Bilateral adnexal tumors commonly present as multicystic masses with nodular areas, excrescences, and papillary projections on their inner surface.	Complex papillary architecture characterized by epithelial projections with irregular contours resulting in the formation of multicellular structures.
Aggressiveness and proliferation rates	Medium-high aggressiveness behavior and moderate proliferation rates.	Less aggressive compared to HGSC, but more aggressive than LGSC. Proliferation rates vary depending on tumor grade and histological characteristics.	Less aggressive compared to other subtypes. Proliferation rates vary based on tumor grade and histological features.	Low aggressive clinical course and low proliferation rate.	Most common and aggressive subtype. Early dissemination and high rates of recurrence.
Genetic aberrations and marker expression	Mutations in *ARID1A/B*, *SMARCA4*, *ERBB2*, *PIK3CA*, *AKT2*, *PTEN*, *KRAS*, *PPP2R1A*	Mutations in *PTEN*, *ARID1A*, *CTNNB1*, *KRAS/BRAF*, *PIK3CA*; aberrant expression of β-catenin, estrogen and progesterone receptors.	Mutations in *KRAS*, *TP53 PIK3CA/PTEN*, *ARID1A*, *BRAF*, *CTNNB1/APC*, elevated levels of CEA, CA-19-9, and REG4.	Mutations in *KRAS*, *BRAF*, *NRAS*, *ERBB2*, *PI3KCA*, *FFAR1*, *USP9X*, and *EIF1AX.*	Mutations in *TP53* and *BRCA1/2*; HER2 amplification.

*HGSC*, high-grade serous carcinoma; *LGSC*, low-grade serous carcinoma.

## Data Availability

Data sharing not applicable.

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
