# Peer review of "Biomarkers in Ovarian Cancer: Towards Personalized Medicine"

_proteomes, 2024, doi:10.3390/proteomes12010008_

Round 1

Reviewer 1 Report

Comments and Suggestions for Authors

This paper should not be accepted for publication in its present form. It could be possibly accepted for publication after major revision. The authors may find the following comments useful for their future work.

-      General Comment: An expert in English should review the paper. In particular, the following expressions should be reviewed:

1) In page 1, line 11: The term “life expectations” should be changed to “life expectancy”.

2) In page 1, line 24: The word female should be deleted (gynecological malignancies are by definition female).

3) In page 1, line 37: The terms “malignant body” and “malignant bodies” should be replaced by the terms “malignant tumor” and “malignant tumors”, respectively.

4) In page 1, line 41: What does “clinical sector” mean?

5) In page 2, line 48: “depicts heightened levels” should be changed to e.g. “contains increased levels”.

6) In page 2, line 58: “Counterparty” should be deleted or replaced by a more appropriate word.

7) In page 2, line 78: The term “prognostic inference” should be changed to “prognosis”.

Other Comments:

-        Page 1, line 18: “In this review…” should be changed to “In this narrative review…”.

-        Page 1, line 24: Ovarian cancer does not always “affect” the fallopian tubes and the peritoneum.

-        Page 1, line 38: The terms “functionality, and physiology” should be changed. Ovarian cancer is by definition a pathologic, not a physiologic entity.  

-        Page 1, line 39: the terms “mesenchymal” and “immunoreactive” are rarely used in clinical practice.

-        Page 2, lines 47-48: VEGF-A is a factor of angiogenesis; hence, “genes relevant to angiogenesis and VEGF-A” should be changed to “genes relevant to angiogenesis”.

-        Page 2, lines 58-62: Radiotherapy is in general contraindicated in ovarian cancer; only very rarely some radiotherapeutic approaches may be considered as therapeutic options in selected cases. Instead, the authors should have mentioned HIPEC among current therapeutic modalities.

-        Page 3, lines 110-119: The different clinical implications between germ-line and somatic mutations of BRCA1 and BRCA2 in ovarian tumors should be discussed.

-        Pages 4-7: Section 3 should be re-written. In the text, the authors should mention the type of sample analyzed in each study (e.g. tumor, peripheral blood, plasma, serum, ascetic fluid etc.). The Table should be divided into panels according to the type of samples analyzed in each study. Furthermore, the authors should discuss the strengths and weaknesses of different studies, e.g. the limitations of analyses in cell lines as opposed to those in clinical samples.

-        Page 7, line 258: As mentioned earlier, radiotherapy is in general contraindicated in ovarian cancer.

-        Pages 4-7: Section 4 should be re-written. The authors should mention the current status of development and applications for each drug-delivery approach; i.e. animal studies, in vitro studies, clinical Phase I, II or III studies etc. Furthermore, though the relatively recent development of using PARP-inhibitors in the treatment of ovarian cancer has been a paradigm shift and a milestone in personalized medicine, it has not been even mentioned in this paper!

Comments on the Quality of English Language

General Comment: An expert in English should review the paper. In particular, the following expressions should be reviewed:

1) In page 1, line 11: The term “life expectations” should be changed to “life expectancy”.

2) In page 1, line 24: The word female should be deleted (gynecological malignancies are by definition female).

3) In page 1, line 37: The terms “malignant body” and “malignant bodies” should be replaced by the terms “malignant tumor” and “malignant tumors”, respectively.

4) In page 1, line 41: What does “clinical sector” mean?

5) In page 2, line 48: “depicts heightened levels” should be changed to e.g. “contains increased levels”.

6) In page 2, line 58: “Counterparty” should be deleted or replaced by a more appropriate word.

7) In page 2, line 78: The term “prognostic inference” should be changed to “prognosis”.

Reviewer 2 Report

Comments and Suggestions for Authors

The review article by Carlos Lopez-Portugues and colleagues summarizes the current knowledge of the developing field of the ovarian cancer biomarkers. The topic of this article is of high importance and could be of interest to the clinical and research community. However, the manuscript does not provide sufficient breadth and depth of the current literature analysis. Multiple review articles are cited instead of the primary research studies, and there are numerous factual inaccuracies. The writing style is confusing, verbose, and lacks a systematic approach. The current or potential use for already established and emerging biomarkers is not well described, leaving the reader disappointed. An extensive discussion of the liposomal nanoparticle technology does not fit well with the topic of the review. Finally, the strengths and weaknesses of different approaches to biomarker discovery, as well as the most promising clinically relevant biomarkers were not thoroughly discussed.

Examples of inadequate descriptions or factual errors:

Lines 37 – 41: Molecular and histological subtypes are described in the same sentence, which is confusing. Accurate classification should be provided, and the description of rare subtypes including clear cell and mucinous carcinomas.

Lines 46 – 49: Again, the confusion with various classifications intermixed. Angiogenesis genes and VEGF-A are not examples of proliferation markers.

Lines 58 – 63: Radiation therapy is rarely used in advanced OC patients, and mostly for pain control.

Comments on the Quality of English Language

The manuscript would benefit from editing for clarity and grammar.

Reviewer 3 Report

Comments and Suggestions for Authors

Abstract: ‘Deathliest’ should read ‘deadliest’ instead. Some of the points made in the abstract are rather disjointed and require further clarification e.g. the statement regarding screening technologies uncovering potential treatment molecules. Perhaps this reads better in the text of the review, but should be paraphrased in the abstract for clarity.

p.1 l.26: “The World Health Organisation (www.who.int) classifies OC into three main groups: epithelial (95% of cases), germ cell and sex-cord-stromal [3]” the internet link should not be to the general WHO site, but to the page specifically dealing with epidemiology of OC. The WHO site should be used as a reference instead of merely a link in parentheses. The citation #3 here is relevant, but not from the WHO, which is the main point of the sentence, and should be used as a supplementary to the appropriate WHO citation. Since the percentage for epithelial ovarian cancer is provided at 95%, the other two categories’ percentages should also be provided from the same source. If this is not possible, different sources may be required.

p.1 l.31: “(data source: 31 GLOBOCAN 2020, International Agency for Research on Cancer).” Similar to the use of the WHO website, this should be cited as a reference, and  not merely in parentheses.

p.1 l.38: The categorisation of epithelial OC is unnecessarily convoluted here. There should be mention of the major five sero-histotypes, and any extraneous categorisation separated. As it currently reads, there are nine categories mentioned of which the latter five are ‘most frequently used’, which is not particularly meaningful or intuitive to the reader: “proliferative, differentiated, mesenchymal, immunoreactive, mucinous carcinoma, endometrioid carcinoma, clear cell carcinoma, high-grade serous carcinoma (HGSOC), and low-grade serous carcinoma, being those last five subtypes the most frequently used”. The following statistics are useful, but ultimately could be better presented e.g. using a pie chart in a figure.

p.2 l.46: Similar to the comment above, mention of biomarkers is useful in-text, however could be better represented e.g. as a table that clearly shows which biomarkers are related to different subtypes, with a column for the appropriate references. Additionally, the provision of a table would allow presentation of more comprehensive data, which is still incomplete in this article.

p.2 l.58: What is meant by ‘counterparty’ in this sentence? “Counterparty, for more advanced stages…” Formatting of this (long) paragraph could be improved to clearly delineate sections on treatment options e.g. i) resection, ii) chemotherapy, iii) radiotherapy, iv) targeted therapies, v) novel therapies etc. A figure to illustrate the different methods on a cellular / molecular level would also be useful.

p.2 l.90: “CA-125, also known as carbohydrate antigen 125.” The abbreviation CA-125 is already used earlier. In the case of use of abbreviations, the full name and abbreviation should always be given in full on first instance of use (p.20 l.51), and the abbreviation can be used subsequently. As a title header, both may be used here (but the instance of first use needs to be modified). Generally, use the following format instead: Carbohydrate Antigen 125 (CA-125). Same point for modification of the subsequent bullet points.

p.3 l.95: When mentioning specificity and sensitivity of tests (as well as other screening metrics, such as PPV and NPV), the values should be provided. This applies for all instances of these statistical mentions throughout the manuscript. The subsequent statement on p.2 l.126 “Despite the great number of markers available for OC diagnosis and prognosis, most of them do not show sufficiently high sensitivity (73.6%±16.8%) nor specificity (76.7%±10.4%) on their own.” requires citations; a generalised sensitivity and specificity statistic may be useful, however, should be presented with further depth on individual marker statistics. Perhaps a table would suit this information better, as this is not a single test for a single / a set of markers, and they should all be treated individually.

p.3 l.117: “Individuals inheriting either a mutated BRCA1 or BRCA2 gene have approximately a 50% chance of passing the mutation to offspring, thus increasing the risk of the descendants having OC [37,38].” This is a rather odd statement to make here, as it is a generalised statement for the inheritance of any allelic mutation being passed down to progeny. It would be more relevant to discuss the statistical correlation of BRCA1/2 mutations with breast / OC, especially in a temporal rate (survival rates would be appropriate here), and then discuss the impact on heritability of 1) mutated allele and 2) disease.

p.4 Fig. 1: A previous statement made in the abstract (as well as subsequently in p.4 l.168, p.5 l.266 etc.) was that screening techniques may give rise to discovery of treatment options. However, the two arms seem to be independent / unrelated in the figure. Perhaps a bit of elaboration in the figure legend would help clarify the utility of both and if they are independent or interdependent.

p.5 l.220:  This paragraph presents some relevant data regarding up/downregulated markers in OC vs. controls. However, caution should be made when remarking that control of these markers can result in management or treatment of OC, as there is not a causative pathway established for all molecules. It may be correct to make this assertion regarding Septin-2 (reference 55), as there is phenotypic change regarding degrease of tumour proliferation, but the other molecules require paraphrasing, as correlation here does not show causation. Overall, this section summarises the information in the literature at a superficial level; the section could be improved by objectively applying a meta-analytical approach to presenting this data – consider listing major findings that are common to many / most papers, as well as critiquing the general fallacies in them (such as there being too many identified molecules without any established causative pathway, leading to poor resolution and impossibility of interpretation of such vast numbers of markers).

p.6 Table 1: Is there a reason ‘mesenchymal’ in the second row is underlined and hyperlinked?

Overall language requires some improvement as there are noticeable grammatical and linguistic errors that obfuscate the meaning of some sentences.

The review suffers from poor structure, making information available throughout the essay, but difficult to access and digest. Perhaps re-working a clearer heading / subheading format would help the review become more readable.

The conclusion section summarises the essay well, however, is rather generic in its findings. A direct critique of the fallacies / insufficiencies of the current evidence, as well as measures taken to address them would be relevant throughout the essay, and deserves a highlight in the conclusion section, setting the tone that the evidence is yet preliminary, and caution should be taken to not over-interpret existing literature.

Referencing format of bibliography seems rather odd, in Spanish, and has inconsistent formatting. Consider revising with the use of a citation software.

Comments on the Quality of English Language

Overall language requires some improvement as there are noticeable grammatical and linguistic errors that obfuscate the meaning of some sentences.

Referencing format of bibliography seems rather odd, in Spanish, and has inconsistent formatting. Consider revising with the use of a citation software.

Round 2

Reviewer 2 Report

Comments and Suggestions for Authors

The revised manuscript is significantly re-written and improved.

Comments on the Quality of English Language

The quality of writing is significantly improved in the revised version.